# Natural Products from Singapore Soil-Derived *Streptomycetaceae* Family and Evaluation of Their Biological Activities

**DOI:** 10.3390/molecules28155832

**Published:** 2023-08-02

**Authors:** Elaine-Jinfeng Chin, Kuan-Chieh Ching, Zann Y. Tan, Mario Wibowo, Chung-Yan Leong, Lay-Kien Yang, Veronica W. P. Ng, Deborah C. S. Seow, Yoganathan Kanagasundaram, Siew-Bee Ng

**Affiliations:** Singapore Institute of Food and Biotechnology Innovation (SIFBI), Agency for Science, Technology and Research (A*STAR), Singapore 138673, Singapore; elainec@sifbi.a-star.edu.sg (E.-J.C.); chingkc@sifbi.a-star.edu.sg (K.-C.C.); tanyq@sifbi.a-star.edu.sg (Z.Y.T.); mario_wibowo@sifbi.a-star.edu.sg (M.W.); leongcy@sifbi.a-star.edu.sg (C.-Y.L.); yanglk@sifbi.a-star.edu.sg (L.-K.Y.); ngwp@sifbi.a-star.edu.sg (V.W.P.N.); seowcs@sifbi.a-star.edu.sg (D.C.S.S.)

**Keywords:** actinobacteria, antimicrobial, *Staphylococcus aureus*, *Streptomycetaceae*, tetronomycin, natural products

## Abstract

Natural products have long been used as a source of antimicrobial agents against various microorganisms. Actinobacteria are a group of bacteria best known to produce a wide variety of bioactive secondary metabolites, including many antimicrobial agents. In this study, four actinobacterial strains found in Singapore terrestrial soil were investigated as potential sources of new antimicrobial compounds. Large-scale cultivation, chemical, and biological investigation led to the isolation of a previously undescribed tetronomycin A (**1**) that demonstrated inhibitory activities against both Gram-positive bacteria *Staphylococcus aureus* (SA) and methicillin-resistant *Staphylococcus aureus* (MRSA) (i.e., MIC_90_ of 2–4 μM and MBC_90_ of 9–12 μM), and several known antimicrobial compounds, namely nonactin, monactin, dinactin, 4E-deacetylchromomycin A3, chromomycin A2, soyasaponin II, lysolipin I, tetronomycin, and naphthomevalin. Tetronomycin showed a two- to six-fold increase in antibacterial activity (i.e., MIC_90_ and MBC_90_ of 1–2 μM) as compared to tetronomycin A (**1**), indicating the presence of an oxy-methyl group at the C-27 position is important for antibacterial activity.

## 1. Introduction

Natural products have been significant in providing the groundwork for the development and advancement of antibiotics since ancient times. Since the discovery of penicillin in 1928, antimicrobial agents linked to antibiotics were mainly isolated from natural sources, such as plants and microorganisms [1,2]. Antimicrobial resistance has been one of the major concerns worldwide. The proliferation of drug-resistant pathogens, which have developed new mechanisms of resistance, poses an ongoing threat to our ability to combat common infections due to antibiotics losing effectiveness. This has resulted in challenging-to-treat infections that can lead to fatal outcomes. Thus, it is crucial to search for other potential antimicrobial agents from natural sources such as actinobacteria [3,4]. Actinobacteria play a pivotal role within the microbial community, as they are recognized as a vital source of innovative bioactive compounds. Approximately 45% of bioactive compounds obtained from microbes were produced by actinobacteria [5,6,7]. Given their extensive biotechnological applications, this group of microorganisms has consistently captivated the interest of chemists, pharmaceutical companies, and various other researchers, making it a compelling subject of study.

Our research team has been actively engaged in a continuous screening endeavor aimed at identifying secondary metabolites derived from actinobacteria, which has the potential to inhibit pathogenic microorganisms, such as *Staphylococcus aureus* (SA) and methicillin-resistant *Staphylococcus aureus* (MRSA) [8,9]. Infections caused by these pathogens are common in both community-acquired and hospital-acquired settings. Among common staphylococcal bacteria, *Staphylococcus aureus* (SA) stands out as particularly dangerous. These Gram-positive, coccus-shaped (spherical) bacteria are not only responsible for skin infections, but also pose a threat by causing pneumonia, cardiovascular related infections, osteomyelitis, and a range of potentially severe infections [10]. One of the reasons why SA is a threat to the society is its ability to develop resistance to antibiotics. MRSA is a well-known example of antibiotic-resistant SA that can be difficult to treat because of their developed mechanisms to evade the effects of many antibiotics commonly used to treat bacterial infections. Actinobacterial-derived drugs have been instrumental in treating various diseases such as microbial and protozoal infections, cancer, and severe inflammations [11]. The diversity and abundance of bioactive compounds produced by actinobacteria make them a valuable resource for the exploration and advancement of drug discovery and development. While there have been previous studies on actinomycetes-derived antibiotics, actinobacteria remains relatively under-explored in the context of Singapore soil. Secondly, although antibiotics have been isolated from actinomycetes in the past, the threat of antibiotic resistance continues to grow, necessitating the constant search for new agents. By focusing on this specific microbial community, we contribute to the ongoing efforts to address the challenges posed by antibiotic resistance and provide valuable insights into the untapped resources of Singapore’s soil ecosystem [4,12].

The objective of our work, therefore, aims to discover new bioactive compounds from actinobacteria strains isolated from Singapore soil with antimicrobial activity. As part of our on-going screening campaign for new bioactive compounds, four actinobacterial strains (A1099, A1174, A1301, and A2461) from our in-house Natural Organisms Library isolated from terrestrial soil in Singapore were grown in five different liquid media [13]. In this study, we report the isolation and characterization of nine known and one new (**1**) natural compounds from these four microbial strains, along with the evaluation of their antimicrobial effect as well as their cytotoxicity against several cancer cell lines.

## 2. Results and Discussion

### 2.1. Phylogenetic Analysis and Molecular Identification of Actinobacteria Isolates

A total of four actinobacterial strains isolated from terrestrial soil in Singapore were molecularly identified via sequencing of the 16S rDNA gene region. This was followed by a nucleotide BLAST search against the NCBI 16S rRNA database with the aligned 16S rRNA gene sequences of A1099, A1174, A1301, and A2461. The neighbor-joining analysis method using a bootstrapped analysis of 1000 replicates of four actinobacteria strains and their closely related species from Genbank database was utilized to investigate their phylogenetic similarity (Figure 1). Our results revealed that strain A1099 shared 99.85% sequence identity (E-value = 0.0) to *Streptomyces badius* with accession number MN966861.1; A1174 shared 99.34% sequence identity (E-value = 0.0) to *Kitasatospora arboriphila* with accession number EU100404.1; A1301 shared 100% sequence identity (E-value = 0.0) to *Streptomyces chattanoogensis* with accession number KM573812.1; and A2461 with 99.64% sequence identity (E-value = 0.0) to *Streptomyces aculeolatus* with accession number MG190783.1. The results show that they are strains from the family of *Streptomycetaceae*. In addition, the genotypic and phenotypic characteristics of genera *Streptomyces* and *Kitasatospora* are difficult to differentiate. They are known to be closely related as shown in the phylogenetic tree in Figure 1 and looking morphologically similar (Figure 2). Morphologically, actinobacteria resemble fungi because of their elongated cells that branch into filaments or hyphae (Figure 2). It is known that these hyphae can be distinguished from fungal hyphae based on size with actinobacteria hyphae being smaller than fungal hyphae [14,15]. Previous studies have shown that actinobacteria from the *Streptomycetaceae* family are exceptional antibiotic producers. They are known to produce various bioactive compounds with antimicrobial activities [16,17].

### 2.2. Preliminary Screening of Actinobacteria Isolates

Culture medium has a great effect not only on microbe growth, but also on metabolism. Studies have shown that the carbon–nitrogen ratio, salinity, and presence of metal ions play a regulatory role in determining the extent and pattern of secondary metabolite production. Typically, culture media primarily consists of carbon and nitrogen sources. The carbon source not only serves as the fundamental building block for biomass and provides energy for microorganisms, but also supplies carbon units to produce secondary metabolites. Similarly, the nitrogen source is essential for synthesizing vital proteins and nucleic acids as well as providing nitrogen-containing units for secondary metabolites [18]. Hence, microorganisms cultured in different medium composition can exhibit differently adapted metabolism, which will produce differential biosynthesis of specialized metabolites.

In this study, 20 extracts were generated from the fermentation of the four actinobacterial strains in five different media. The extracts were tested for their inhibitory effects against a series of microbial pathogens, including *Klebsiella aerogenes* (KA), *Pseudomonas aeruginosa* (PA), *Staphylococcus aureus* (SA), *Candida albicans* (CA), and *Aspergillus fumigatus* (AF). In addition, their cytotoxic activity towards the human lung carcinoma A549, pancreatic cancer MIA, and pancreatic cancer PANC-1 cell lines was evaluated. Antimicrobial and cytotoxic primary screenings of the extracts derived from four actinobacterial strains exhibited biological activities against at least one of the tested microbial pathogens or cancer cell lines (Figure 3 and Appendix A). The findings revealed noticeable variations in the biological activities expressed by crude extracts derived from the same actinobacterial strain when cultivated in different growth media. Previous studies have reported that deliberate manipulation of different media substance is a tactic to identify a favorable growth regime, which enhances the diversity of metabolites and the production of bioactive secondary metabolites. This approach, commonly referred to as the OSMAC (one strain, many compounds) approach, has been reported to yield promising results [19,20,21]. As observed in the results shown in Figure 3 and Appendix A strain A1099 showed activity against SA and CA when fermented in CA02LB, CA07LB, and CA08LB, whereas A1174 showed activity against SA and A549 when fermented in CA08LB. As for A1301 in CA10LB, it showed the most significant results with activity against three microbial pathogens—SA, CA, AF—and three cancer cell lines—A549, MIA, PANC-1—while in CA07LB, the extract was active against SA, CA, A549, MIA, PANC-1. Extracts generated in CA02LB was only active against AF, whereas activity was only observed against SA in extracts from CA08LB and CA09LB. Strain A2461 cultured in CA08LB and CA10LB exhibited activity against SA. Further analysis of the primary bioassay screening showed that A1099 strain, when fermented in CA08LB, exhibited the highest percentage of inhibition against SA and CA compared to CA02LB and CA07LB (Appendix A). For A2461 strain, antibacterial activity against SA was more pronounced in terms of percentage of inhibition when fermented in CA10LB in comparison with CA08LB (Appendix A). Thus, CA08LB extract of A1099 and CA10LB extract of A2461 were selected as the preferred extracts for further investigation. CA08LB was selected as a preferred media to ferment strain A1174 because this was the only extract that showed antibacterial activity against SA (Figure 3). In addition, strain A1301 fermented in CA10LB was selected for further investigation due to its broad spectrum of antimicrobial activity against three microbial pathogens as well as cytotoxic activity (Figure 3). From this study, CA08LB and CA10LB were found to be the optimal media for production of bioactive compounds. Notably, these media contain relatively higher amount of carbohydrates as compared to the others. Carbohydrates play a crucial role in fermentation processes as a source of energy for microorganisms. During fermentation, carbohydrates are broken down into simpler compounds, such as sugars, which are then metabolized by microorganisms to produce various fermentation products, including alcohol, organic acids, and gases. Studies have shown that fermentation with high amount of carbohydrate substance helped enhance production of bioactive compounds [22].

### 2.3. Isolation and Structural Elucidation of Bioactive Compounds

Selected extracts from the 4 actinobacterial strains were subjected to large-scale bioassay guided fractionation. This study was performed not only to confirm the antimicrobial activities of the active metabolites, but also to expand our in-house natural compounds library [13]. Large-scale cultivation of 4 actinobacterial strains and purification of active metabolites from their extracts led to the identification of numerous known metabolites as summarized in Table 1. These known metabolites, namely nonactin, monactin, dinactin, 4E-deacetylchromomycin A3, chromomycin A2, soyasaponin II, lysolipin I, tetronomycin, and naphthomevalin (Figure 4) were identified and characterized using high-resolution mass spectroscopy (HRMS) and nuclear magnetic resonance (NMR) analyses and spectroscopic data comparison with the literature values [23,24,25,26,27,28,29,30,31,32,33]. In addition, one new natural product, tetronomycin A (**1**), was also isolated from the extract derived from *Streptomyces* sp. A2461 fermented in CA10LB (Table 1). Examples of structure elucidation of tetronomycin and tetronomycin A (**1**) are presented below.

Tetronomycin was isolated as one of the bioactive compounds from the extract derived from *Streptomyces aculeolatus* A2461 (Figure 5). The structure of tetronomycin consists of four methyl, ten methylene, thirteen methine, and six non-protonated carbons was confirmed through detailed analyses of the ^1^H and 2D NMR spectra (Appendix A). ^1^H-^1^H COSY and HMBC correlations were deduced as shown in Figure 6 to establish the core skeleton of tetronomycin. In addition, the identity of tetronomycin was also confirmed through comparing the ^1^H NMR spectrum of tetronomycin in CDCl_3_ with literature data [23,24]. The assignment of the ^13^C chemical shifts of tetronomycin was conducted based on correlations observed in HSQC and HMBC NMR spectra of tetronomycin (Table 2 and Appendix A).

Tetronomycin A (**1**) (Figure 5) was isolated as a white amorphous powder. The molecular formula was established as C_33_H_48_O_8_ based on HR-ESIMS analysis. The structure of **1** was established based on NMR data comparison with those of tetronomycin. Following the literature review, the ^1^H NMR spectrum of **1** (Appendix A) was found to be similar to that of tetronomycin except for the absence of a methoxy group in **1** [23,24]. Compound **1** was found to possess the same core structure as tetronomycin with the loss of a methyl functional group on the oxy-methyl group at C-27 position as indicated by the missing proton singlet at δ_H_ 3.33 (Figure 5 and Table 2). In addition, there is a slight difference between the ^1^H chemical shifts of H-27 and H-26 in the ^1^H NMR spectra of **1** and that of tetronomycin. The chemical shift of H-27 shifted downfield, changing from δ_H_ 3.37 to δ_H_ 3.82 while the chemical shift of H-26 moved upfield, changing from δ_H_ 4.15 to δ_H_ 4.03. The loss of one methyl group caused a change in the chemical environment around C-26 and C-27 positions and, thus, the change in their chemical shifts. In addition, the core skeleton of tetronomycin A consists of a tetronic acid, a tetrahydrofuran, and a tetrahydropyran fragments, which was established based on ^1^H-^1^H COSY and HMBC correlations (Figure 6 and Appendix A). Unfortunately, due to the low yield of compound **1**, not all ^13^C NMR shifts could be assigned from ^13^C NMR experiment. Therefore, the ^13^C NMR chemical shifts were obtained from HSQC and HMBC spectra (Appendix A). The structure of **1** was very similar with those of the known tetronomycin, suggesting they were biosynthetically related. Thus, based on spectroscopic data comparison and biosynthetic consideration, the relative configurations for **1** were proposed to be the same as those in tetronomycin. Notably, the sign of optical rotation of **1** was the opposite to that of tetronomycin in this study (${\left[\alpha \right]}_{D}^{23}$ + 111, c 0.0003, MeOH), which was in accordance with previously reported data [23]. Although a comparison of optical rotation signs of similar structures had been used in many studies, it has been shown that the sign of the optical rotation is an unreliable indicator of stereochemistry determination in natural products, and the signs of the optical rotations of two compounds can be opposite regardless of their identical configurations [34,35].

### 2.4. Chemical Structural Data of Tetronomycin A (**1**)

The UV spectra and HRESIMS spectra of **1** and 1D and 2D NMR spectra of tetronomycin and **1** are provided in Appendix A.

**1**: White amorphous powders; ${\left[\alpha \right]}_{D}^{23}$-61 (c 0.001, MeOH); UV (MeCN/H_2_O) λ_max_ (%) 222 (100%), 296 (26%) nm; (+)-HRESIMS: *m*/*z* 595.3249 [M + Na]^+^ (calcd for C_33_H_48_NaO_8_, 595.3247); ^1^H and ^13^C NMR data, see Table 2.

### 2.5. Antimicrobial and Cytotoxic Activities of Compounds Isolated from the 4 Actinobacterial Strains

Tetronomycin A (**1**), tetronomycin and eight other known compounds isolated from A1099, A1174, A1301, and A2461 were subjected to antimicrobial and cytotoxicity dose-response testing against a panel of five microbial pathogens, *K. aerogenes* (KA)*, P. aeruginosa* (PA)*, S. aureus* (SA)*, C. albicans* (CA), *and A. fumigatus* (AF), and three cancer cell lines, A549, MIA PaCa-2, and PANC-1. Table 3 shows the antimicrobial and cytotoxicity activities of the 8 known compounds from A1099, A1174, A1301, and A2461. Bioactivity testing results showed that nonactin, monactin, and dinactin isolated from A1099 exhibited antimicrobial activity towards the Gram-positive bacteria SA and antifungal activity against CA. These compounds are from a family of naturally occurring cyclic ionophores known as the macrotetrolide antibiotics [29,31]. Similarly, glycosylated tricyclic aureolic polyketides 4E-deacetylchromomycin A3 and chromomycin A2 isolated from A1174 exhibited activity against SA, consistent with what was reported in the literature [36], and known compounds lysolipin I and soyasaponin II isolated from A1301 showed activity against SA. Interestingly, soyasaponin II demonstrated similar antimicrobial activity as chromomycin compounds (i.e., MIC_90_ of 2–4 µM) and better antimicrobial activity than nonactin, monactin, and dinactin. Soyasaponin II is a complex oleanane triterpenoid that was reported to have hepatoprotective, antiviral (i.e., anti-herpes simplex virus activity), and cardiovascular protective activity [37]. However, no antimicrobial activity against SA was previously reported, even though a similar analog, soyasaponin I, was reported to exhibit antimicrobial activity against *E. coli.* and CA [37]. Soyasaponins are a group of triterpenoids commonly found on soybeans, which is part of CA10LB media component. This suggested that the soyasaponin II isolated in this work was not produced by the actinobacterial strain but part of the media component instead. In addition, lysolipin I also showed antifungal activity against CA and AF. The most potent compound is lysolipin I as it showed sub-micromolar antimicrobial activity against SA, CA, and AF (i.e., 0.01–0.9 µM) while naphthomevalin did not exhibit any antibacterial activity, which is consistent with what was reported in the literature [38]. In comparison with the respective positive controls tested, all compounds isolated showed less potent activity except lysolipin I. No bioactivity was observed in these compounds against Gram-negative bacteria (KA and PA) (Appendix A). In addition, the eight known compounds exhibited cytotoxicity activity towards all three cancer cell lines as shown in Table 3 and Appendix A. These bioactivity findings are consistent with reports on the bioactivity of these known compounds [32,39,40]. However, the known compounds of previous studies were isolated from different *Streptomyces* species not investigated in this study. *Streptomyces* species have a vast genetic diversity, and each strain may possess unique biosynthetic capabilities. As a result, different strains of *Streptomyces* can produce a variety of secondary metabolites with similar or overlapping bioactivities.

Previous studies have shown that tetronomycin exhibited potent antibacterial activity against drug-resistant strains [24]. Thus, tetronomycin A (**1**) and tetronomycin isolated from A2461-CA10LB were subjected to additional screening to investigate their potential activity against the drug-resistant bacteria MRSA (Table 4). Figure 7 and Figure 8 show the dose-response inhibition curves of the compounds and their IC_90_ values for SA and MRSA as well as the IC_50_ values for the cytotoxicity against the three cancer cell lines, respectively. As shown in Table 4, both 1 and tetronomycin showed potent antibacterial activities against SA as well as MRSA. Tetronomycin was more potent (i.e., minimal inhibitory concentration (MIC_90_) of 0.8 µM and minimal bactericidal concentration (MBC_90_) of 1.7 µM) against SA than 1 (i.e., MIC_90_ of 2.2 µM and MBC_90_ of 9.2 µM). Similarly for MRSA, tetronomycin (i.e., MIC_90_ of 0.9 µM and MBC_90_ of 2.1 µM) was more active compared to 1 (i.e., MIC_90_ of 3.9 µM and MBC_90_ of 11.8 µM). This two- to six-fold decrease in antibacterial activity was observed in 1 as compared to tetronomycin when the oxy-methyl group at C-27 position was changed to a hydroxy group. This may indicate the importance of oxy-methyl group as a pharmacologically active group. In comparison with positive control, tetronomycin showed similar bioactivity with vancomycin hydrochloride with one slightly higher bioactivity of MBC 1.7 µM against SA. These two compounds were inactive against KA, PA, CA, and AF (dose-response curves were shown in Appendix A).

Tetronomycin was first isolated from a cultured broth of *Streptomyces* sp. in 1982 [23]. It is a polycyclic polyether compound. Recently, Kimishima et al. reported the bioactivity of tetronomycin and their semi-synthetic analogues [24]. The research group investigated acyl derivatives of tetronomycin and other derivatives that did not possess an *exo*-methylene group on the tetronic acid moiety. Acyl derivatives were reported to have similar antimicrobial activity profile as tetronomycin, but the derivatives exhibited less potent antimicrobial activity than tetronomycin while the exo-methylene moiety in tetronomycin was crucial for its antimicrobial activity. Interestingly, our A2461 *Streptomyces aculeolatus* produces tetronic acid compounds (i.e., 1 and tetronomycin), which was not reported in the literature. On the other hand, *Streptomyces aculeolatus* was reported to produce naphthoquinone derivatives, such as aculeolatins A-D and 2,5,7-trihydroxy-3, 6-dimethylnaphthalene-1,4-dione, which shared similar core structures as one of our isolated compounds, naphthomevalin. This strain was also reported to produce compounds that demonstrated antimalarial, anti-tuberculosis, antibacterial, and weak cytotoxicity activities [41]. The findings of our study not only serve to further demonstrate the actinobacteria as a prolific natural source for antimicrobial drug discovery, but also significantly contribute to enriching the structural diversity of microbial natural products. By identifying new bioactive compound from actinobacteria strains isolated from Singapore soil, we expand the repertoire of potential antimicrobial agents and enhance our understanding of the wide range of structural variations that microbial natural products can exhibit.

### 2.6. Effects of Growth Media on Production of Bioactive Compounds

To unravel the effects of growth media on actinobacteria for their potential to enhance bioactive metabolite biosynthesis as well as bioactivity of the crude extracts, the abundance of the isolated bioactive compounds produced by the four actinobacteria strains were compared in different media as presented in Figure 9. From our primary screening results, SA activity was only observed in A1174 fermented in CA08LB (Figure 3). This is consistent with the abundance of bioactive compounds, 4E-deacetylchromomycin A3 (*m*/*z* 1141.5033) and chromomycin A2 (*m*/*z* 1211.5472) found in extracts derived from different media. These chromomycin analogues were only found in extract derived from A1174 fermented in CA08LB but not found in extracts derived from other media (Figure 9B). In Figure 9C, a higher abundance of lysolipin I (*m*/*z* 598.1121) was observed in A1301 extract fermented in CA10LB as compared to other media. This possibly led to a broader spectrum of antimicrobial activity against three pathogens, SA, CA, and AF observed in CA10LB extract (Figure 3). On the other hand, SA activity was observed in extracts derived from A2461 grown in CA08LB and CA10LB, but not in extracts grown in other media (Figure 3). Interestingly, no SA activity was observed in the extract derived from A2461 grown in CA07LB even though both tetronomycin (*m*/*z* 609.3408) and tetronomycin A (**1**) (*m*/*z* 595.3249) were present. The observed bioactivity in the CA08LB and CA10LB extracts is likely due to a higher abundance of tetronomycin in the CA08LB and CA10LB extracts compared to the CA07LB extract (Figure 9D). Lastly, the activity observed in extracts derived from A1099 fermented in different media did not show any correlation with the abundance of nonactin (*m*/*z* 737.4514), monactin (*m*/*z* 751.4650), and dinactin (*m*/*z* 765.4801) found in different extracts (Figure 9A). Even though the abundance of these macrotetrolides were the highest in CA10LB extract, no activity was observed as shown in Figure 3. However, CA08LB extract was selected for further isolation and purification work because it showed the highest percentage of inhibition against SA and CA in our primary screening results. This finding further exemplified the OSMAC method as a promising strategy for diversification of secondary metabolite production.

## 3. Materials and Methods

### 3.1. Molecular Identification and Phylogenetic Analysis of Actinobacteria Isolates

Actinobacteria strains used in this study were obtained from the Natural Product Library, which were initially isolated from terrestrial soils in Singapore’s nature parks [13]. These strains were derived from soil samples collected at Singapore’s Bukit Batok Nature Park and Kent Ridge Park and were isolated using two specific types of agar media. The isolation media utilized were humic acid-vitamin agar and arginine-glycerol-salt agar. Reference stock cultures stored at −80 °C were revived and sub-cultured on Bennet’s Agar (Oxoid, Hampshire, UK), followed by incubation at 28 °C for 24 h. Manufacturers’ protocol from DNeasy PowerSoil Kit (Qiagen, Hilden, Germany) was followed and conducted to isolate genomic DNA of strains of interest. The extracted DNA were quantified using NanoDrop2000 spectrophotometer (ThermoFisher Scientific, San Diego, CA, USA). Amplification of 16S rDNA genes of interest were carried out using universal 16S primers 27F (5′—AGA GTT TGA TCC TGG CTC AG—3′) and 1492R (5′—TAC GGY TAC CTT GTT ACG ACT T—3′) [42,43]. The total PCR amplification reaction mixture of 20 µL consists of 2.0 µL of 10× PCR buffer with 20 mM MgCl_2_, 2.0 µL of 2 mM dNTPs, 0.2 µL of Dream Taq polymerase (ThermoFisher Scientific, Waltham, MA, USA), 1.0 µL of 10 µM of each primer, and 1.0 μL of purified DNA templates. A negative control and non-template were included in the run. The PCR amplifications were performed using Applied Biosystems ProFlex Thermocycler (ThermoFisher Scientific, Waltham, MA, USA) with the following thermal cycling profile conditions of initial denaturation at 95 °C for 5 min; further denaturation of 30 cycles at 95 °C for 30 s each; annealing at 60 °C for 30 s; followed by initial extension at 72 °C for 1 min and a final extension at 72 °C for 5 min.

PCR products were electrophoresed on 1% agarose gels (1× TAE buffer, 1 g agarose gel) stained with SYBR safe DNA gel stain (ThermoFisher Scientific, Waltham, MA, USA). The agarose gels were visualized on a ChemiDoc™ MP Imaging System (Bio-Rad, Hercules, CA, USA). PCR products then underwent purification using MEGA quick-spin total fragment DNA purification kit (iNtRON Biotechnology, Seongnam, Republic of Korea) following manufacturer’s instructions. Purified PCR products were then sent for bi-directional sequencing services (1st BASE, Singapore) using the mentioned primer pair. Alignment and analysis of the sequences was done using Benchling and BLAST [National Center for Biotechnology Information (NCBI)]. The 4 actinobacteria strains were aligned using ClustalW with the 16S rRNA regions of closely related strains retrieved from Gen-Bank databases. The neighbor-joining tree algorithm method was utilized to determine the genetic relationship between the strains. In order to construct the phylogenetic tree, MEGA 11.0 software (Mega, PA, USA) was employed using a bootstrapped analysis of 1000 replicates [44]. DNA sequences of A1099, A1174, A1301, A2461 have been uploaded to the GenBank database of NCBI under the accession number OR177839, OR177840, OR177841, and OR177842 respectively.

### 3.2. Fermentation and Extraction of Actinobacterial Crude Extracts

Four actinobacterial strains were selected for extracts generation following phylogenetic analysis. A volume of 5 mL SV2 media (for 1 L, add 1 g calcium carbonate (Sigma-Aldrich, St. Louis, MO, USA), 15 g glucose (1st BASE, Singapore), 15 g glycerol (VWR, Radnor, PA, USA), and 15 g soya peptone (Oxoid, Hampshire, UK), pH adjusted to 7.0) was used to culture strains at 28 °C for 3 days under constant agitation at 200 rpm to generate a seed culture. The seed cultures were then inoculated in a 1:20 volume into five in-house liquid media (CA02LB, CA07LB, CA08LB, CA09LB, and CA10LB) as shown in Table 5. These media have been formulated and optimized by the Natural Product Library group at SIFBI for actinobacteria secondary metabolites production. The cultures were incubated for 9 days at 28 °C in the dark with shaking at 200 rpm. Following incubation, the cultures were lyophilized. The dried cultures underwent extraction using methanol (MeOH) and were subsequently filtered through Whatman Grade 4 filter paper. MeOH was then evaporated under reduced pressure to generate the crude extract.

### 3.3. Biological Assays

The 20 extracts generated from the fermentation of the 4 strains in 5 media were first subjected through a primary screening campaign. These extracts were screened for anti-microbial activity against selected bacterial and fungal strains, which were *Klebsiella aerogenes,* KA (ATCC^®^ 13048™); *Pseudomonas aeruginosa,* PA (ATCC^®^ 9027™); *Staphylococcus aureus* Rosenbach, SA (ATCC^®^ 25923™); *Candida albicans,* CA (ATCC^®^ 10231™), and *Aspergillus fumigatus,* AF (ATCC^®^ 46645™). Primary screening for the cytotoxic effects of the extracts was also done on A549 human lung carcinoma cells (ATCC^®^ CCL-185™) and two pancreatic cancer cell lines, which were MIA PaCa-2 (ATCC^®^ CCL-1420™) and PANC-1 cells (ATCC^®^ CCL-1469™). Primary screening was performed in triplicate at a single concentration of 100 µg/mL to determine its percentage of inhibition activity of crude extracts. The criteria of active hits were antimicrobial effect and cytotoxic activity with an average growth inhibition ≥ 80%. Following a bioactivity-guided primary testing strategy for compound isolation, selected active hits from extracts with desired bioactivity were then subjected to scale-up fermentation for isolation of active compounds.

Dose-response testing of the isolated compounds for the antimicrobial and cytotoxicity testing was performed in triplicates using a sixteen-point, 2-fold serial dilution assay format with a starting assay concentration of 100 μM. For the anti-microbial bioassays, a modified version of the microbroth dilution method established in alignment with the Clinical Laboratory Standards Institute (CLSI) guidelines was performed to investigate the minimum inhibition concentration (MIC) and the minimum bactericidal/fungicidal concentration (MBC/MFC) of the isolated compounds. Bacterial MIC testing was done by incubating the isolated compounds with 5.5 × 10^5^ cfu/mL of bacterial cells at 37 °C for 24 h. For fungal MIC testing against CA the compounds were incubated with 2.5 × 10^3^ cfu/mL and incubated at 25 °C for 48 h. Whereas for MIC testing against AF, the compounds were incubated with the fungal cells seeded at a concentration of 2.5 × 10^4^ spores/mL, followed by incubation at 25 °C for 72 h. OD_600_ absorbance readings of the cultures were performed after incubation to determine the inhibitory effect of the compounds on the microbes. To further study the potential bactericidal and fungicidal effects of the compounds, 5 µL of the treated culture was inoculated into freshly dispensed media in microtiter plates. The microtiter plates were then incubated using the same condition for the respective microbes, followed by OD_600_ measurement. Vancomycin hydrochloride and amphotericin B (Sigma-Aldrich, St. Louis, MO, USA) were utilized as the standard inhibitor controls for the antibacterial and antifungal assays, respectively. The isolated compounds were also subjected to cytotoxicity testing against human carcinoma cell lines. The human carcinoma cells were seeded at a density of 3.3 × 10^4^ cells/mL, followed by treatment with the compounds at 37 °C for 72 h under 5% CO_2_ condition. PrestoBlue™ cell viability reagent (ThermoFisher Scientific, Waltham, MA, USA) was used for quantification of cytotoxic effects via fluorescence reading at an emission of 590 nm and an excitation of 560 nm. Puromycin (Sigma-Aldrich, St. Louis, MO, USA) were used as the standard inhibitor controls for the cytotoxicity assays. Bioassay results were analysed using GraphPad Prism 8 software (GraphPad, San Diego, CA, USA) to determine the respective IC_50_ and IC_90_ values.

### 3.4. Natural Product Extraction, Compound Isolation, and Structure Elucidation

The 4 actinobacterial strains underwent large scale fermentation of 4 L in their respective selected media of interest, which were CA08LB and CA10LB. Following incubation, the cultures were lyophilized. The dried cultures were extracted with MeOH and filtered through filter paper (Whatman Grade 4). This was followed by removal of MeOH in vacuo to obtain the crude extracts from strains A1099 (weight of 10.20 g), A1174 (weight of 15.03 g), A1301 (weight of 6.36 g), and A2461 (weight of 0.98 g). A volume of 2.5 mL MeOH was added to each of the extracts to generate a saturated solution of extract. The saturated solution was then subjected to C_18_ reversed-phase preparative HPLC purification. Solvent A was water + 0.1% formic acid, solvent B was acetonitrile + 0.1% formic acid, and flow rate was 30–52 mL/min. These conditions were the same for purification of compounds from all 4 extracts. For A1099, the extract was fractionated by the following condition (gradient conditions: isocratic condition of 15% B for 5 min, 30 mL/min; followed by linear increment of flow rate to 52 mL/min over 5 min; 15–42% B over 28 min, 52 mL/min; 42–100% B over 24 min; and isocratic condition of 100% B for 10 min) to give 19.8 mg of nonactin, 20.5 mg of monactin, and 47.8 mg of dinactin. For A1174, the extract was fractionated by the following condition (gradient conditions: isocratic condition of 25% B for 5 min, 30 mL/min; followed by linear increment of flow rate to 52 mL/min over 5 min; 25–60% B over 42 min, 52 mL/min; 60–100% B over 10 min; and isocratic condition of 100% B for 10 min) to give 3.6 mg of 4E-deacetylchromomycin A3 and 5.0 mg of chromomycin A2. For A1301, the extract was fractionated by the following condition (gradient conditions: isocratic condition of 15% B for 5 min, 30 mL/min; followed by linear increment of flow rate to 52 mL/min over 5 min; 15–32% B over 15 min, 52 mL/min; 32–65% B over 35 min; 65–100% B over 2 min; and isocratic condition of 100% B for 10 min) to give 0.8 mg of soyasaponin II and 2.4 mg of lysolipin I. For A2461, the extract was fractionated by the following condition (gradient conditions: isocratic condition of 20% B for 5 min, 30 mL/min; followed by linear increment of flow rate to 52 mL/min over 5 min; 20–45% B over 10 min, 52 mL/min; 45–85% B over 40 min; 85–100% B over 2 min; and isocratic condition of 100% B for 10 min) to give 0.8 mg of **1**, 1.0 mg of tetronomycin, and 0.8 mg of naphthomevalin. Known metabolites, namely nonactin, monactin, dinactin, 4E-deacetylchromomycin A3, chromomycin A2, soyasaponin II, lysolipin I, tetronomycin, and naphthomevalin were confirmed by comparison of NMR and ESI-HRMS data with the literature values [23,24,25,26,27,28,29,30,31].

### 3.5. General Chemistry Experimental Procedures

Several instruments were used to characterize the chemical properties of the compounds; for example, P-2000 digital polarimeter (JASCO) was used to measure the specific rotations of the compounds and Bruker DRX-400 NMR spectrometer with 5-mm BBI (1H, G-COSY, multiplicity-edited G-HSQC, and G-HMBC spectra) probe heads equipped with z-gradients and Cryoprobe was utilized to collect NMR spectra of the compounds. The ^1^H chemical shifts were referenced to the residual solvent peaks for CDCl_3_ at δ_H_ 7.26 ppm and (CD_3_)_2_CO at δ_H_ 2.05 and δ_C_ 29.8 ppm, respectively. C_18_ reversed-phase preparative HPLC purification was conducted using Agilent 1260 Infinity Preparative-Scale LC/MS Purification System coupled to Agilent 6130B single quadrupole mass spectrometer with Agilent 5 Prep C18 column (100 × 30 mm, 5 µm). HPLC-MS analyses were conducted using Agilent UHPLC 1290 Infinity coupled to Agilent 6540 accurate-mass quadrupole time-of-flight (QTOF) mass spectrometer and an ESI source. Gradient elution that starts from 98% water with 0.1% formic acid to 100% acetonitrile with 0.1% formic acid over 8.6 min along with an Acquity UPLC BEH C_18_ (2.1 × 50 mm, 1.7 µm) column at a flow rate of 0.5 mL/min was used. The operating parameters for QTOF were the same as previously reported [45].

## 4. Conclusions

A series of actinobacterial strains were isolated from a soil sample collected in Singapore and were found to produce several known antimicrobial compounds, namely nonactin, monactin, dinactin, 4E-deacetylchromomycin A3, chromomycin A2, soyasaponin II, lysolipin I, tetronomycin, and naphthomevalin and a newly discovered tetronomycin A derivative (**1**) that exhibited antibacterial activity against SA and MRSA, with MIC_90_ values ranging from 2 to 4 μM and MBC_90_ values ranging from 9 to 12 μM. In addition, this study showed the importance of an oxy-methyl group at C-27 position of tetronomycin for antibacterial activity. This report also further demonstrated actinobacteria as a potential natural source for antimicrobial drug discovery and provided better understanding on tetronomycins as potent antibacterial agents. In addition, the findings also demonstrated OSMAC method as a possible strategy to enhance the production of a diverse bioactive secondary metabolites in actinobacteria.

The discovery of antimicrobial compounds in this study warrants future investigation into the specific biochemical interactions through which a substance produces its pharmacological effect (mechanism of action studies). Moreover, the discovery of tetronomycin A (**1**) could lead to medicinal chemistry research to generate compounds libraries for structure activity relationships (SAR) and chemical biology studies, owing to the presence of the secondary hydroxy moiety at C-27 (i.e., incorporating ester or carbamate moieties). However, the relatively low yield of pure compounds obtained in this study could be the limiting factor for future works. Thus, larger scale isolation studies would be necessary to obtain higher quantity of compounds.

## Figures and Tables

**Figure 1 molecules-28-05832-f001:**
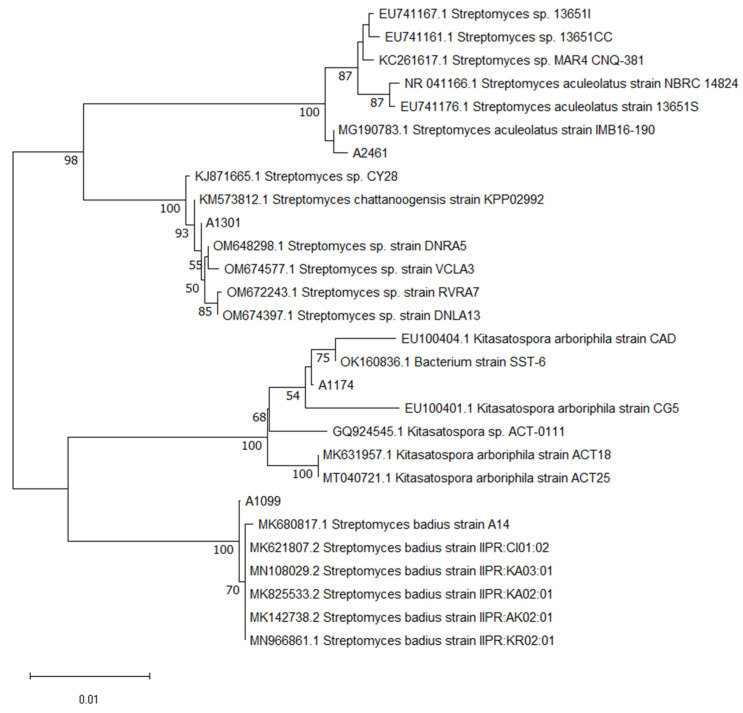
Phylogenetic tree showing the evolutionary relationship between strains A1099, A1174, A1301, and A2461 and other type species of the family *Streptomycetaceae*. Neighbor-joining phylogenetic tree was constructed based on 16S rRNA gene sequence showing the relationship between strains A1099, A1174, A1301, A2461 and representatives or related actinobacteria strains retrieved from the GenBank with their respective accession numbers. Bootstrap values greater than 50% are shown at the number on the branches nodes that were analyzed based on 1000 replicates. Bar, 0.01 substitutions per nucleotide position.

**Figure 2 molecules-28-05832-f002:**
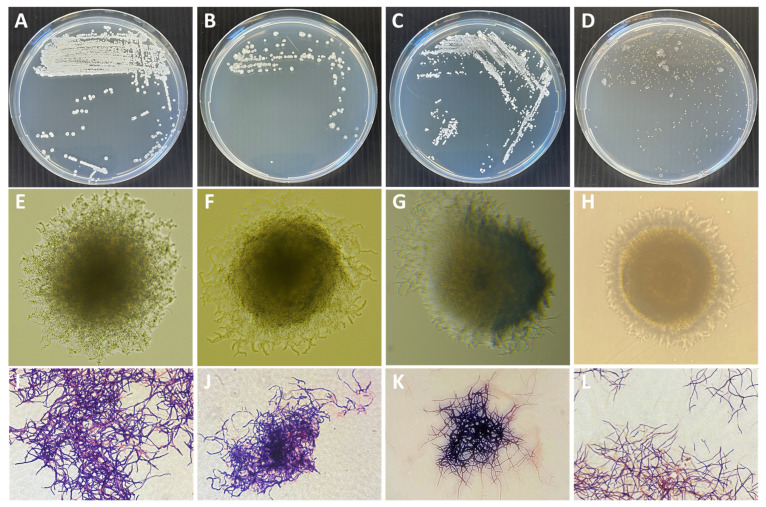
Visual images of macroscopic plate image of strains (**A**) A1099, (**B**) A1174, (**C**) A1301, and (**D**) A2461. Visual images of colony morphology (magnification: 200×) (**E**) A1099, (**F**) A1174, (**G**) A1301, and (**H**) A2461. Gram staining of actinobacteria (magnification: 1000×) (**I**) A1099, (**J**) A1174, (**K**) A1301, and (**L**) A2461.

**Figure 3 molecules-28-05832-f003:**
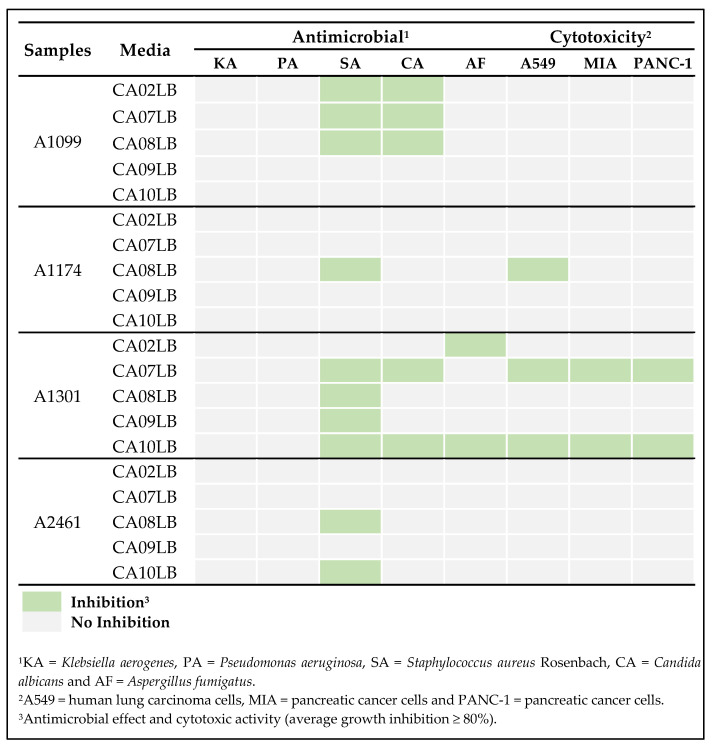
Antimicrobial and cytotoxicity primary screening results of 4 actinobacteria strains grown in 5 different growth media.

**Figure 4 molecules-28-05832-f004:**
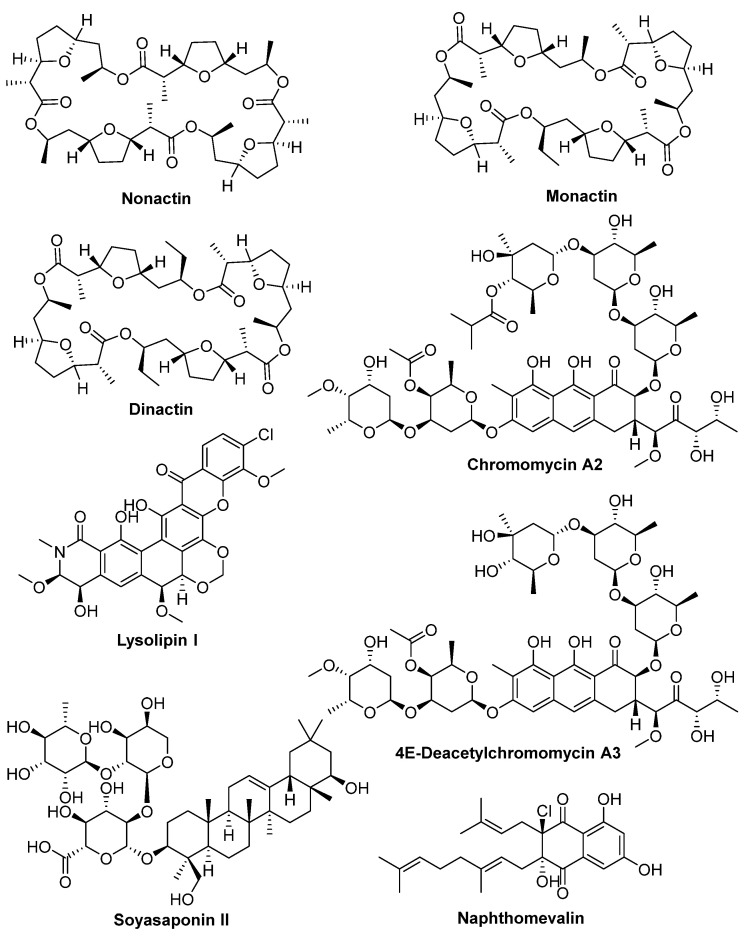
Chemical structures of nonactin, monactin, dinactin, 4E-deacetylchromomycin A3, chromomycin A2, soyasaponin II, lysolipin I, tetronomycin, and naphthomevalin.

**Figure 5 molecules-28-05832-f005:**
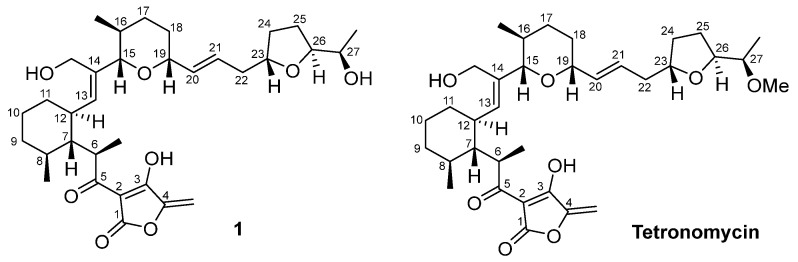
Chemical structures of **1** and tetronomycin.

**Figure 6 molecules-28-05832-f006:**
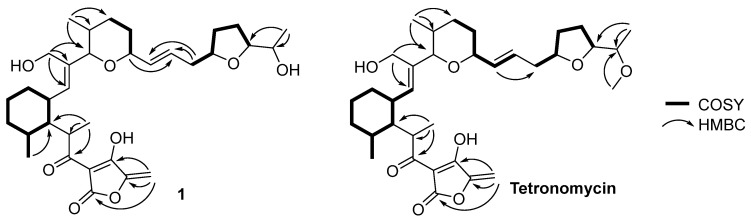
Selected COSY and HMBC correlations for **1** and tetronomycin.

**Figure 7 molecules-28-05832-f007:**
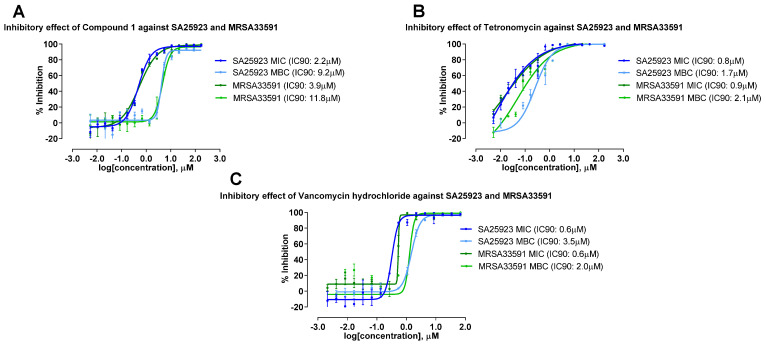
Dose response inhibition curves against *Staphylococcus aureus* Rosenbach (SA25923) and methicillin-resistant *Staphylococcus aureus* subsp. aureus Rosenbach (MRSA33591). (**A**) Tetronomycin A (**1**), (**B**) Tetronomycin, and (**C**) Vancomycin hydrochloride.

**Figure 8 molecules-28-05832-f008:**
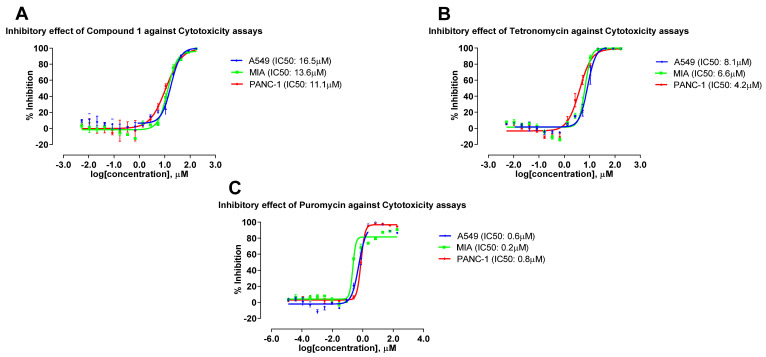
Dose response curves against A549 human lung carcinoma cells, and two pancreatic cancer cell lines MIA PaCa-2 and PANC-1 cells. (**A**) Tetronomycin A (**1**), (**B**) Tetronomycin, and (**C**) Puromycin.

**Figure 9 molecules-28-05832-f009:**
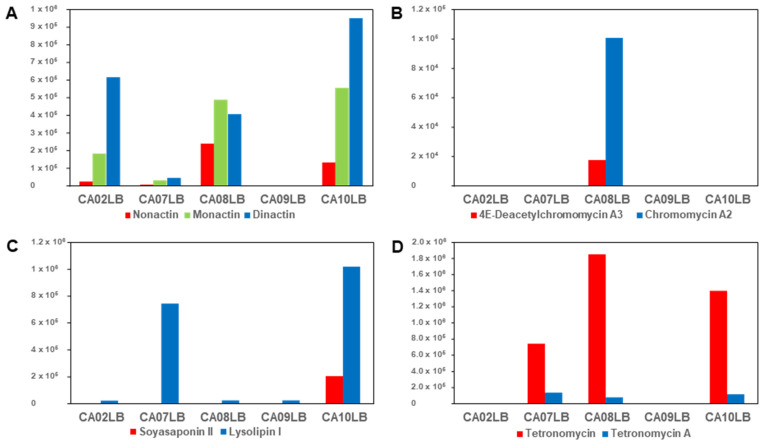
Comparison of relative abundance (peak area) of various bioactive compounds from the four actinobacterial strains (A1099, A1174, A1301, and A2461) fermented in different media (CA02LB, CA07LB, CA08LB, CA09LB, and CA10LB). (**A**) *Streptomyces* sp. A1099, (**B**) *Kitasatospora* sp. A1174, (**C**) *Streptomyces* sp. A1301, (**D**) *Streptomyces* sp. A2461.

**Table 1 molecules-28-05832-t001:** Compounds identified by comparison of their NMR data with the literature values [23,24,25,26,27,28,29,30,31,32,33].

Strain	Media	Compounds Confirmation
*Streptomyces* sp. A1099	CA08LB	Nonactin, monactin, dinactin
*Kitasatospora* sp. A1174	CA08LB	4E-Deacetylchromomycin A3, chromomycin A2
*Streptomyces* sp. A1301	CA10LB	Soyasaponin II, lysolipin I
*Streptomyces* sp. A2461	CA10LB	Tetronomycin A (**1**), tetronomycin and naphthomevalin

**Table 2 molecules-28-05832-t002:** ^1^H (400 MHz) NMR data of **1** and tetronomycin in (CD_3_)_2_CO.

	1		Tetronomycin	
Pos.	^13^C, Type ^1^	^1^H, Mult. (*J* = Hz)	^13^C, Type ^1^	^1^H, Mult. (*J* = Hz)
1	179.7, C	-	180.5, C	-
2	n.d., C	-	n.d., C	-
3	182.6, C	-	182.3, C	-
4	155.2, C	-	155.6, C	-
4-CH_2_	85.9, CH_2_	4.55, d (1.0); 4.93, d (1.0)	85.9, CH_2_	4.54, d (1.0); 4.93, d (1.0)
5	201.4, C	-	201.3, C	-
6	43.3, CH	3.73, m	43.3, CH	3.81, m
6-Me	8.6, CH_3_	0.94, d (7.0)	9.0, CH_3_	0.98, d (7.1)
7	48.5, CH	1.83, m	48.5, CH	1.81, m
8	33.1, CH	1.48, m	33.1, CH	1.45, m
8-Me	20.0, CH_3_	1.14, m	20.1, CH_3_	1.14, m
9	36.3, CH_2_	1.08, m; 1.64, m	36.1, CH_2_	1.06, m; 1.64, m
10	25.9, CH_2_	1.30, m; 1.59, m	25.9, CH_2_	1.27, m; 1.58, m
11	35.4, CH_2_	1.01, m, 1.44, m	35.7, CH_2_	1.00, m, 1.45, m
12	36.4, CH	2.54, m	36.5, CH	2.55, m
13	141.6, CH	5.10, d (10.1)	141.5, CH	5.10, d (10.1)
14	132.5, C	-	n.d., C	-
14-CH_2_	56.5, CH_2_	3.83, m; 4.14, m	56.5, CH_2_	3.84, m; 4.19, m
15	91.7, CH	3.19, m	91.7, CH	3.20, m
16	34.5, CH	1.40, m	34.4, CH	1.39, m
16-Me	18.2, CH_3_	0.58, d (6.6)	18.3, CH_3_	0.58, d (6.8)
17	32.6, CH_2_	1.23, m; 1.80, m	32.7, CH_2_	1.22, m; 1.80, m
18	32.1, CH_2_	1.48, m; 1.61, m	32.0, CH_2_	1.46, m; 1.60, m
19	80.0, CH	3.80, m	80.0, CH	3.79, m
20	132.2, CH	5.57, dd (8.6, 15.1)	132.8, CH	5.54, dd (8.7, 15.6)
21	135.4, CH	6.19, m	n.d., CH	6.14, m
22	39.8, CH_2_	2.19, m; 2.39, m	40.1, CH_2_	2.06, m; 2.38, m
23	78.7, CH	4.10, m	78.9, CH	4.11, m
24	32.3, CH_2_	1.58, m; 2.12, m	32.2, CH_2_	1.60, m; 2.12, m
25	26.8, CH_2_	1.72, m; 1.92, m	27.7, CH_2_	1.65, m; 1.96, m
26	82.2, CH	4.03, m	80.7, CH	4.15, m
27	68.3, CH	3.82, m	78.8, CH	3.37, dq (2.4, 6.4)
27-Me	16.5, CH_3_	0.97, d (6.3)	11.1, CH_3_	0.95, d (6.4)
27-OMe	-	-	57.0, CH_3_	3.33, s

^1^ Assignments based on HSQC and HMBC spectra, and comparison with the literature values of tetronomycin [23,24]. Chemical shifts (δ) in ppm. n.d. = not determined.

**Table 3 molecules-28-05832-t003:** Biological activities of positive controls and 8 known compounds isolated from A1099-CA08LB, A1174-CA08LB, A1301-CA10LB, A2461-CA10LB.

Sample	Media	Compound	Antimicrobial (µM) ^1^						Cytotoxicity (µM) ^2^		
			SA		CA		AF		A549	MIA	PANC-1
			MIC_90_	MBC_90_	MIC_90_	MFC_90_	MIC_90_	MFC_90_	IC_50_	IC_50_	IC_50_
A1099	CA08LB	Nonactin	49.2	64.9	38.1	-	-	-	10.1	2.3	2.9
		Monactin	7.9	-	1.1	8.2	-	-	0.8	0.1	0.1
		Dinactin	4.7	-	1.3	4.0	-	-	1.2	0.7	0.3
A1174	CA08LB	4E-Deacetylchromomycin A3	2.9	13.4	-	-	-	-	1.7	1.9	3.0
		Chromomycin A2	3.1	3.8	-	-	-	-	0.3	0.5	0.4
A1301	CA10LB	Lysolipin I	0.01	NT	0.1	NT	0.9	NT	0.1	0.2	0.3
		Soyasaponin II	2.8	2.4	-	-	-	-	2.1	3.1	2.1
A2461	CA10LB	Naphthomevalin	-	-	-	-	-	-	3.7	6.4	9.0
Positive Controls		Vancomycin hydrochloride	0.6	3.5							
		Amphotericin B			0.1	0.2	0.5	1.7			
		Puromycin							0.6	0.2	0.8

^1^ SA = *Staphylococcus aureus* Rosenbach, CA = *Candida albicans*, and AF = *Aspergillus fumigatus*. (–) Compounds show no inhibition for MIC_90_ and MBC_90_/MFC_90._ ^2^ A549 = human lung carcinoma cells, MIA = pancreatic cancer cells, and PANC-1 = pancreatic cancer cells. NT indicates that compound of interest was not tested.

**Table 4 molecules-28-05832-t004:** Biological activities of positive controls, tetronomycin A (**1**), and tetronomycin from A2491-CA10LB.

Compounds	Antimicrobial (µM) ^1^				Cytotoxicity (µM) ^2^		
	SA25923		MRSA33591		A549	MIA	PANC-1
	MIC_90_	MBC_90_	MIC_90_	MBC_90_	IC_50_	IC_50_	IC_50_
Tetronomycin A (**1**)	2.2	9.2	3.9	11.8	16.5	13.6	11.1
Tetronomycin	0.8	1.7	0.9	2.1	8.1	6.6	4.2
Vancomycin hydrochloride	0.6	3.5	0.6	2.0			
Puromycin					0.6	0.2	0.8

^1^ SA = *Staphylococcus aureus* Rosenbach, MRSA = Methicillin-resistant *Staphylococcus aureus*. ^2^ A549 = human lung carcinoma cells, MIA = pancreatic cancer cells, and PANC-1 = pancreatic cancer cells.

**Table 5 molecules-28-05832-t005:** Composition of the five media that were used in this study.

Components	Media (per L)				
	CA02LB	CA07LB	CA08LB	CA09LB	CA10LB
Lab-lemco, Oxoid LP0029	-	-	-	10 g	-
Cane molasses	-	-	20 g	-	-
Cottonseed flour	-	-	25 g	-	-
Glucose	-	-	15 g	20 g	-
Glycerol	-	15 g	-	3 g	-
Mannitol	20 g	-	-	-	-
Oatmeal	-	30 g	-	-	-
Soluble starch	-	-	40 g	-	20 g
Soybean meal	20 g	-	-	-	15 g
Yeast extract	-	5 g	-	4 g	-
CaCO_3_	-	-	8 g	-	-
KH_2_PO_4_	-	5 g	-	-	3 g
Na_2_HPO_4_·12H_2_O	-	5 g	-	-	2 g
MgCl_2_·6H_2_O	-	1 g	-	-	-
MgSO_4_·7H_2_O	-	-	-	-	0.5 g
Trace salt sol ^1^	-	-	-	-	1 mL
pH	7.5	Natural	7.2	7.0	7.2

^1^ Trace salt solution consists of 0.2 g each of FeSO_4_·7H_2_O, MnCl_2_·4H_2_O, ZnSO_4_·7H_2_O, CuSO_4_·5H_2_O, and CoCl_2_·2H_2_O in 100 mL.

## Data Availability

The data supporting the finding in this study are contained within the article or Appendix A.

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
