# Peer review of "Natural Products from Singapore Soil-Derived Streptomycetaceae Family and Evaluation of Their Biological Activities"

_molecules, 2023, doi:10.3390/molecules28155832_

Round 1
Reviewer 1 Report
The article presents the results obtained by investigating 4 strains of antibiotic-producing actinobacteria. In my opinion, the research, the quality of the figures and the expression in English are excellent, so I recommend the publication in its current form. Only after Material and Method, on page 14, is there a comma that needs to be cut. Also, in the bibliography, indexes 20 and 26 must be corrected (some text appears written in capital letters).
Author Response
Reviewer 1
- The article presents the results obtained by investigating 4 strains of antibiotic-producing actinobacteria. In my opinion, the research, the quality of the figures and the expression in English are excellent, so I recommend the publication in its current form.
We thank the reviewer for the positive comments.
- Only after Material and Method, on page 14, is there a comma that needs to be cut. Also, in the bibliography, indexes 20 and 26 must be corrected (some text appears written in capital letters).
We thank the reviewer for pointing this out. The changes have been made on page 14, indexes 21, 23, 28, 29 and other minor corrections.

Reviewer 2 Report
In the presented work, Chin et al., describe the isolation and characterization of a new compound, tetronomycin A (1) and other known metabolites from Soil-derived Streptomycetaceae. Although the work has a good appeal, there is a fundamental lack data in compound 1, the most relevant compound in the manuscript. No 13C NMR data was presented for compound 1, moreover no data or explanations were given as to how the stereochemistry of this compound (as presented in Figure 5) was established. Without the mentioned core relevant data, the manuscript cannot be accepted for publication in Molecules. The authors are therefore encouraged to present the requested data and explanations.
Please see further specific comments on the manuscript in the attached pdf file.

The quality of the English language just requires minor revisions as mentioned to the authors in the specific comments of the attached pdf file.
Author Response
- In the presented work, Chin et al., describe the isolation and characterization of a new compound, tetronomycin A (1) and other known metabolites from Soil-derived Streptomycetaceae. Although the work has a good appeal, there is a fundamental lack in data for compound 1, the most relevant compound in the manuscript. No 13C NMR data was presented for compound 1.
We thank the reviewer for the comments. We have performed 13C NMR experiment for compound 1, unfortunately the low yield of compound 1 led to the situations where not all 13C NMR shifts could be assigned from its 13C NMR spectrum. Hence, the 13C NMR chemical shifts were obtained from HSQC and HMBC spectra as we have mentioned in the original submission. We believe 13C NMR shifts assignment based on 2D NMR (HSQC and HMBC) is scientifically acceptable and is a common practice for natural products with low yields. A few examples of previously published papers reporting 13C NMR chemical shifts based on HSQC and HMBC (without 13C NMR spectrum): J. Nat. Prod. 2015, 78, 12, 2908–2916 and J. Nat. Prod. 2016, 79, 4, 946–953.
To clarify this, we have added a footnote below Table 2 that reads “1Assignments based on HSQC and HMBC spectra, and comparison with literature compound values of tetronomycin”.
- No data or explanations were given as to how the stereochemistry of this compound (as presented in Figure 5) was established.
We thank the reviewer for pointing this out. We have added discussion on the relative configurations determination for compound 1. The paragraph now reads:
“The structure of 1 was very similar with those the known tetronomycin, suggesting they were biosynthetically related. Thus, based on spectroscopic data comparison and bio-synthetic consideration, the relative configurations for 1 were proposed to be the same as those in tetronomycin.”
Of note, consideration of biosynthetic origin has been regarded as a crucial part in structure determination, as it has been nicely highlighted in Nat. Prod. Rep., 2017, 34, 1193; DOI: 10.1039/c7np00025a.
- Page 7 line 186 reads “seven methylene, sixteen methylene”. The authors should correctly identify the number of methylene and methine groups.
We thank the reviewer for pointing this out. The number of methylene and methine groups has been corrected. The sentence now reads “seven methylene, sixteen methine”.
- Pages 7-8, lines 193-197 reads “δC chemical shift of C-2 was not determined because there is no neighboring proton to establish a HMBC correlation with C-2. In addition, the δC chemical shifts of C-14 and C-21 were not determined because no HMBC cross peak between neighboring protons and C-14 as well as 1H-13C HSQC cross peak for H-21 was observed (Figures S7 and S8).” This section rather detracts than add anything relevant to the manuscript and is better deleted from manuscript, because a 13C NMR would be the most appropriate method of assigning chemical shifts to all the carbon signals. Moreover, tetronomycin is a known and published compound, hence no need to go into its detailed structure determination.
We thank the reviewer for the constructive comments. The section has been omitted.
- Page 8 line 199 HR-ESIMS measurement should read HR-ESIMS analysis.
The requested change has been made.
- Page 8 lines 215-216 reads C chemical shift of C-2 was not determined because there is no neighboring proton to establish a HMBC correlation with C-2. Why 13C NMR used in assigning all the carbon chemical shifts in compound 1?
We thank the reviewer for the comments. We have performed 13C NMR experiment for compound 1, unfortunately the low yield of compound 1 led to the situations where not all 13C NMR shifts could be assigned from its 13C NMR spectrum. Hence, the 13C chemical shifts were obtained from HSQC and HMBC spectra as we have mentioned in the original submission. We believe 13C NMR shifts assignment based on 2D NMR (HSQC and HMBC) is scientifically acceptable and is a common practice for natural products with low yields. A few examples of previously published papers reporting 13C NMR chemical shifts based on HSQC and HMBC (without 13C NMR spectrum): J. Nat. Prod. 2015, 78, 12, 2908–2916 and J. Nat. Prod. 2016, 79, 4, 946–953.
To clarify this, we have added a footnote below Table 2 that reads “1Assignments based on HSQC and HMBC spectra, and comparison with literature compound values of tetronomycin”. Moreover we have added these explanation in the paragraph that reads “Unfortunately, due to the low yield of compound 1, not all 13C NMR shifts could be assigned from 13C NMR experiment. Therefore, the 13C NMR chemical shifts were obtained from HSQC and HMBC spectra.”
- How was the stereochemistry in compound 1 established as shown in Figure 5? Please explain.
We have added discussion on the relative configurations determination for compound 1. The paragraph now reads:
“The structure of 1 was very similar with those the known tetronomycin, suggesting they were biosynthetically related. Thus, based on spectroscopic data comparison and bio-synthetic consideration, the relative configurations for 1 were proposed to be the same as those in tetronomycin.”
Of note, consideration of biosynthetic origin has been regarded as a crucial part in structure determination, as it has been highlighted in Nat. Prod. Rep., 2017, 34, 1193; DOI: 10.1039/c7np00025a.
- Page 9 line 233: the calculated [M+Na]+ should be C33H48NaO8 and not C33H49O8 as stated.
The calculated [M+Na]+ has been amended.
- Page 9 line 232 quotes [α]D -61 as the optical rotation of 1. Can the authors explain why compound 1 would have an opposite optical rotation to that of tetronomycin, when they claim that both compounds have exactly the same stereochemistry (Figure 5) and the only difference between the two compounds is the loss of the methoxy group at C-27 of compound 1 (Note that the optical rotation of tetronomycin has been reported as [α]D +125.5 and [α]D +80.5 in references 18 and 19 respectively). This is an additional reason why the authors must explain in detail how they established the stereochemistry of compound 1.
We thank the reviewer for the comments. Even though comparison of optical rotation signs of similar structures had been used in many studies, it has been shown that the sign of the optical rotation is an unreliable indicator of stereochemistry determination in natural products. Previous studies have shown that a slight variation in the structure (i.e. additional of a methyl group) could change the sign of optical rotation, hence the signs of the optical rotations of two compounds can be opposite regardless of their identical configurations. A couple of references of these studies are: J. Nat. Prod. 2011, 74, 4, 707–711 and J. Nat. Prod. 2012, 75, 10, 1792–1797. Therefore, we proposed the stereochemistry in compound 1 to be the same as in tetronomycin based on their similar spectroscopic data and biosynthetic consideration. Moreover, compound 1 and tetronomycin were isolated from the same bacterial strain, which further supported their biosynthetic relationships.
Of note, consideration of biosynthetic origin has been regarded as a crucial part in structure determination, as it has been highlighted in Nat. Prod. Rep., 2017, 34, 1193; DOI: 10.1039/c7np00025a
We have added a discussion text about optical rotations in the manuscript. The text reads “Notably, the sign of optical rotation of 1 was the opposite to that of tetronomycin. Although comparison of optical rotation signs of similar structures had been used in many studies, it has been shown that the sign of the optical rotation is an unreliable indicator of stereochemistry determination in natural products, and the signs of the optical rotations of two compounds can be opposite regardless of their identical configurations.”
- Page 10 line 240: “against similar antimicrobial panel” of should read “against a panel of” .
The requested change has been made.
- Page 15 lines 409-411: The aim of fermenting the 4 actinobacteria strains in different media compositions . There is no need to state an aim in this section. Moreover, the authors have explained the rationale behind the OSMAC method a couple of times already.
We thank the reviewer for the suggestion. The sentence has been deleted.
- Page 17 lines 504-505 makes mention of the use of Bruker DRX-400 NMR spectrometer with 5-mm. BBO (13C spectra) yet no 13C NMR spectrum is shown in the supplementary data, neither does it seem that any of carbon chemical shifts in tetronomycin and compound 1 were assigned by 13C NMR. Why is this the case?
Thank you for pointing this. We have omitted the “13C spectra” from the materials and method section.
- Page 18 line 525 reads preliminary structure-activity relationship studies. This claim cannot be made as no -activity relationship studi was done in this work, at least, such data has not been presented here. The authors can just mention that the C-27 methoxy group seem to slightly increase the antibacterial potency of tetronomycin in comparison to compound 1.
Thank you for the suggestions. We have omitted “preliminary SAR” in the entire manuscript.
- The first page of this section which contains the list of figures should be totally separated from the pages that start showing the figures. As it stands now, it seems as if Figure 14 also corresponds to the UV spectrum of 1.
The requested change has been made.
- The authors start this section with Figure S1. UV spectrum of 1 and Figure S2. (+)-HRESIMS spectrum for 1 which is appropriate because compound 1 is the most relevant compound in the manuscript, but they then skip to present the 1H NMR of tetronomycin in CDCl3 and (CD3)2CO. This disorganizes the whole data presentation. It would be better for the authors to continue presenting all the NMR data on 1 before moving on to another compound.
The requested change has been made.
- The structures of both compound 1 and tetronomycin should be included in their respective 1H NMR spectra for easy reference.
The requested change has been made.
- Since tetronomycin is a known compound with published NMR data, there is no need to include all its NMR data in the supplementary data. Just including 1H NMR and HSQC data here would be sufficient.
The requested change has been made.

Reviewer 3 Report
The study titled “Natural Products from Singapore Soil-derived Streptomycetceae family and their Biological Activities Evaluation” has been reviewed and found substantial concerns in the present form. There are several antibiotics agents already isolated from various types of actinomycetes. Thus, the novelty of the study is important and major gaps evident in the literature and how the present study mitigates those gaps are required to be highlighted. Also, latest studies sin the introductions (2018-2023) are limited in the introduction. The objectives of the study are not clearly mentioned in the introduction.
There are no details about the identification of the sampling sites, rationale, and sample collection protocols. It is very important to add these details for the reproduction of the study. The bioinformatics analysis of the phylogenetic tree and what tree building program and tree evaluation methods used are not clear. There is no description about the boots strap values and it is not properly interpreted. The morphological features (microscopy) of these fungi along with Figure should be supplemented. The hyphal arrangements and other mycelial features are required.
What is the rationale of the selection of the strain for antibacterial activities? These rae standard MTCC or ATCC?
The large-scale bioassay guided fractionation of the molecules is not clear. What are the major techniques used for the purification of each compound are their justification is important. The detailed purification data and results of each study should be supplemented in supplementary materials. The MIC and MBC with respective to appropriate control experiments are not clear. The control experiments for all eth experiments are also not clear.
The potential limitation of the present study and possibilities of the future works are missing in the present study. There is no proper discussion in comparison with latest studies of the similar kinds.
The novelty and uniqueness of the study are not clear in the conclusion.
The manuscript requires professional proof reading with help of an expert to enhance the quality of overall presentation.
Must be improved. Mentioned in the comments.
Author Response
Reviewer 3
- The study titled “Natural Products from Singapore Soil-derived Streptomycetceae family and their Biological Activities Evaluation” has been reviewed and found substantial concerns in the present form. There are several antibiotics agents already isolated from various types of actinomycetes. Thus, the novelty of the study is important and major gaps evident in the literature and how the present study mitigates those gaps are required to be highlighted. Also, latest studies sin the introductions (2018-2023) are limited in the introduction.
We thank the reviewer for the comments. We acknowledge the concerns raised regarding the novelty of our study, given the existence of several antibiotics agents already isolated from various types of actinomycetes. However, we would like to highlight the unique aspects our study aims to address.
Firstly, while there have been previous studies on actinomycetes-derived antibiotics, actinobacteria remains relatively unexplored in the context of Singapore soil. Our study focuses on soil-derived actinomycetes isolated from Singapore local habitats which offers the potential for the discovery of novel bioactive compounds with diverse chemical structures and biological activities.
Secondly, although antibiotics have been isolated from actinomycetes in the past, the threat of antibiotic resistance continues to grow, necessitating the constant search for new agents. Our study aims to contribute to this urgent need by exploring the potential of the actinobacteria strains from Singapore soil as a source of novel antibiotics. By examining this unique ecological niche, we hope to identify previously unknown bioactive compounds that can help mitigate the challenges posed by antibiotic resistance.
The requested changes have been, more references (2018-2023) have been added to the list.
As the reviewer suggested, we have extended the introduction section to further highlight the importance of our studies. The additional text reads “While there have been previous studies on actinomycetes-derived antibiotics, actinobacteria remains relatively under-explored in the context of Singapore soil. Secondly, although antibiotics have been isolated from actinomycetes in the past, the threat of antibiotic resistance continues to grow, necessitating the constant search for new agents. By focusing on this specific microbial community, we contribute to the ongoing efforts to address the challenges posed by antibiotic resistance and provide valuable insights into the untapped resources of Singapore's soil ecosystem”
- The objectives of the study are not clearly mentioned in the introduction.
We thank the reviewer for the comments. We believe we have mentioned the objective of our study in the original submission. The objective of our work was to discover new bioactive compounds from actinobacteria strains isolated from Singapore soil with antimicrobial activity.
- There are no details about the identification of the sampling sites, rationale, and sample collection protocols. It is very important to add these details for the reproduction of the study.
We thank the reviewer for the comments. The requested change has been made. We have added the respective details of sampling site and collection protocol information in “Materials and Method” in part “3.1 Molecular Identification and Phylogenetic Analysis of Actinobacteria Isolates”. These strains were isolated from soil collected in Singapore’s nature park which are Bukit Batok Nature Park and Kent Ridge Park. The isolation media used were humic acid-vitamin agar and arginine-glycerol-salt agar. These strains were isolated from terrestrial soils in Singapore, with the aim of expanding the Natural Product Library by adding to the collection of strains derived from local habitats.
The additional text reads “Actinobacteria strains used in this study were obtained from the Natural Product Library, which were initially isolated from terrestrial soils in Singapore’s nature parks. These strains were derived from soil samples collected at Singapore's Bukit Batok Nature Park and Kent Ridge Park were isolated using two specific types of agar media. The isolation media utilized were humic acid-vitamin agar and arginine-glycerol-salt agar”
- The bioinformatics analysis of the phylogenetic tree and what tree building program and tree evaluation methods used are not clear. There is no description about the boots strap values and it is not properly interpreted.
We thank the reviewer for the comments. The 4 actinobacteria strains were aligned using ClustalW with the 16S rRNA regions of closely related strains retrieved from Gen-Bank databases. The neighbor-joining tree algorithm method was utilized to determine the genetic relationship between the strains. In order to construct the phylogenetic tree, MEGA 11.0 software (Mega, USA) was employed, using a bootstrapped analysis of 1000 replicates. These explanations are found in “Materials and Methods” in part “3.1 Molecular Identification and Phylogenetic Analysis of Actinobacteria Isolates”.
- The morphological features (microscopy) of these fungi along with Figure should be supplemented. The hyphal arrangements and other mycelial features are required.
We thank the reviewer for the comments. The requested changes have been made. We have added a few more microscopy features of these actinobacteria as shown in Figure 2.
- What is the rationale of the selection of the strain for antibacterial activities? These rae standard MTCC or ATCC?
We thank the reviewer for the question. The strains used for screening for antimicrobial activities are Klebsiella aerogenes, KA (ATCC® 13048™); Pseudomonas aeruginosa, PA (ATCC® 9027™); Staphylococcus aureus Rosenbach, SA (ATCC® 25923™); Candida albicans, CA (ATCC® 10231™) and Aspergillus fumigatus, AF (ATCC® 46645™). These are ATCC strains. The rationale of selection of the strains for antimicrobial screening, are these bacteria strains are part of the six ESKAPE highly virulent and antibiotic resistant bacterial pathogens and two fungal strains were included in the testing to discover antifungal activities.
- The large-scale bioassay guided fractionation of the molecules is not clear. What are the major techniques used for the purification of each compound are their justification is important. The detailed purification data and results of each study should be supplemented in supplementary materials.
We thank the reviewer for the comments. The large-scale bioassay guided fractionation of the molecules for the purification of each compound is done using C18 reversed-phase preparative HPLC purification. Solvent A is water + 0.1% formic acid, solvent B is acetonitrile + 0.1% formic acid and flow rate is 30-52 mL/min. These explanations are found in “Materials and Methods” in part “3.4 Natural Product Extraction, Compound Isolation, and Structure Elucidation”.
- The MIC and MBC with respective to appropriate control experiments are not clear. The control experiments for all eth experiments are also not clear.
We thank the reviewer for the comments. Vancomycin hydrochloride and amphotericin B were utilized as the standard inhibitor controls for the antibacterial and antifungal assays, respectively. Whereas puromycin were used as the standard inhibitor controls for the cytotoxicity assays. These controls information is found in “Materials and Methods” in part “3.3 Biological Assays”. Results of the positive controls are found in Table 3 and 4.
- The potential limitation of the present study and possibilities of the future works are missing in the present study.
We thank the reviewer for the comments. We have added a paragraph for the potential limitation of the present study and possibilities of the future works in the discussion as suggested in the conclusion section. The paragraph reads “The discovery of antimicrobial compounds in this study warrants future investigations on the specific biochemical interactions through which a substance produces its pharmacological effect (mechanism of action studies). Moreover, the discovery of tetronomycin A (1) could lead to medicinal chemistry research to generate compounds libraries for structure activity relationships (SAR) and chemical biology studies, owing to the presence of the secondary hydroxy moiety at C-27 (i.e. incorporating ester or carbamate moieties). However, the relatively low yield of pure compounds obtained in this study could be the limiting factor for future works. Thus, larger scale isolation studies would be necessary to obtain higher quantity of compounds.”
- There is no proper discussion in comparison with latest studies of the similar kinds.
We thank the reviewer for the comments. Discussion and comparisons with recent studies on similar compounds can be found in the “Results and Discussion” section, specifically in part “2.5 Antimicrobial and Cytotoxic Activities of Compounds Isolated from the 4 Actinobacterial Strains”. This section provides a discussion of the antimicrobial and cytotoxic properties of the compounds obtained from the four actinobacterial strains, in relation to findings from other recent research in the field.
- The novelty and uniqueness of the study are not clear in the conclusion.
We thank the reviewer for the comments. We believe the discovery of the new compound, tetronomycin A and its biological activity in this study are unique and novel. This report also further demonstrated actinobacteria as a potential natural source for antimicrobial drug discovery and provided better understanding on tetronomycins as potent antibacterial agents.
- The manuscript requires professional proof reading with help of an expert to enhance the quality of overall presentation.
We thank the reviewer for the comments. English have been checked and proof-read.

Round 2
Reviewer 2 Report
Although the authors have addressed most of the concerns and errors raised in the first round of review, the manuscript still contains some minor errors which must be addressed as detailed below:
(1) Page 8 line 229 should read: “similar with those of the known tetronomycin (“of” is missing in the phrase)
(2) Can the authors explain how they came by “seven methylene, sixteen methine” groups in the molecule as indicated on page 7 line 203? Because one could count ten methylene groups, including the oxymethylene (14-CH2-OH) and olefinic methylene (4-CH2), and even if these two are excluded, one can still count eight methylenes. And it is even more strange how the authors´ have reviewed the manuscript and still counted “sixteen methine” groups because only thirteen methine groups are present in the molecule.
(3) Page 10 line 263 the authors state “[M+Na]+ (calcd for C33H49NaO8, 595.3247)”, when this was corrected for the authors in the first review as C33H48NaO8 (i.e. if the molecular formula of Tetronomycin A (1) as quoted by the authors on page 8 line 212 is “C33H48O8”, then [M+Na]+ should be C33H48NaO8 as corrected for the authors in the previous review and not the “C33H49NaO8” indicated in the revised manuscript).
(4) For reasons of comparison, and to support their argument, the authors should also include in their manuscript the [α]D23 measurement for the tetronomycin isolated in the present work.
Author Response
Reviewer 2
(1) Page 8 line 229 should read: “similar with those of the known tetronomycin (“of” is missing in the phrase)
The suggested change has been made. The phrase now reads “similar with those of the known tetronomycin”.
(2) Can the authors explain how they came by “seven methylene, sixteen methine” groups in the molecule as indicated on page 7 line 203? Because one could count ten methylene groups, including the oxymethylene (14-CH2-OH) and olefinic methylene (4-CH2), and even if these two are excluded, one can still count eight methylenes. And it is even more strange how the authors´ have reviewed the manuscript and still counted “sixteen methine” groups because only thirteen methine groups are present in the molecule.
We thank the reviewer for pointing this out. We overlooked the errors in the previous revision. In the current version of manuscript, we have amended the number of methylene and methine to “ten methylene, thirteen methine”. In addition, we have revised the type of C-22, C-24, and C-25 in Table 2 to CH2.
(3) Page 10 line 263 the authors state “[M+Na]+ (calcd for C33H49NaO8, 595.3247)”, when this was corrected for the authors in the first review as C33H48NaO8 (i.e. if the molecular formula of Tetronomycin A (1) as quoted by the authors on page 8 line 212 is “C33H48O8”, then [M+Na]+ should be C33H48NaO8 as corrected for the authors in the previous review and not the “C33H49NaO8” indicated in the revised manuscript).
The suggested change has been made.
(4) For reasons of comparison, and to support their argument, the authors should also include in their manuscript the [α]D23 measurement for the tetronomycin isolated in the present work.
We thank the reviewer for their comment. The optical measurement for tetronomycin in this study has been added as part of the structure elucidation discussion. The sentence now reads “. Notably, the sign of optical rotation of 1 was the opposite to that of tetronomycin in this study (+111, c 0.0003, MeOH), which was in accordance with previously reported data [23].

Reviewer 3 Report
The authors have addressed the comments from the referees. The manuscript can be accepted for publication
Minor English editing is required
Author Response
We thank reviewer 3 for no comment.